# Study on ESD Protection Circuit by TCAD Simulation and TLP Experiment

**DOI:** 10.3390/mi14030600

**Published:** 2023-03-04

**Authors:** Fuxing Li, Changchun Chai, Yuqian Liu, Yanxing Song, Lei Wang, Yintang Yang

**Affiliations:** Key Laboratory of Ministry of Education for Wide Band-Gap Semiconductor Materials and Devices, School of Microelectronics, Xidian University, Xi’an 710071, China

**Keywords:** ESD protection circuit, TLP, damage mechanism, TCAD simulation

## Abstract

The anti-ESD characteristic of the electronic system is paid more and more attention. Moreover, the on-chip electrostatic discharge (ESD) is necessary for integrated circuits to prevent ESD failures. In this paper, the mixed TCAD model of the ESD protection circuit is built and simulated, and the negative transmission line pulse (TLP) injection damage experiment is carried out on the CD4069UBC chip. The circuit model consists of three-dimensional shallow trench isolation (STI) diode TCAD models and a three-dimensional multi-gate Complementary Metal-Oxide-Semiconductor (CMOS) inverter TCAD model. Moreover, the TCAD modeling is based on a 0.25 μm technology node. Through the transient simulation of the electrothermal coupling, the electrical signal of the input and output nodes of the circuit and the distribution of the electrothermal parameters in the device are obtained. Moreover, by analyzing the simulation results, the physical phenomena and the mechanisms of interference and damage mechanism during TLP injection are explained. The location and type of diode damage in the TLP injection simulation results of the circuit model are consistent with the TLP experiment damage results. The proposed ESD protection circuit model and analysis method are beneficial to ESD robustness prediction and ESD soft damage analysis of IC.

## 1. Introduction

With the development of semiconductor technology, chips are widely used in various fields. Unlike consumer electronics, cost reduction is often the primary consideration in the design process, and chip reliability requirements in some new areas are more important, such as automotive, medical, aerospace, and so on [1,2].

Electrostatic discharge (ESD) protection has always been an important aspect of the reliability of electronic systems [3]. Moreover, on-chip electrostatic discharge (ESD) protection is necessary for integrated circuits to prevent ESD failures [4]. To achieve higher reliability standards, testing requirements for ESD protection are stricter, testing methods are complex and diverse, and testing costs are increasing exponentially [5]. For example, the International Organization for Standardization (ISO) automotive electronic control unit electrostatic discharge test standard requires direct contact discharge tests on all connector pins, which can result in very time-consuming tests that can take up to hundreds of times [6]. Through ESD circuit modeling, multiple physical quantity changes in circuit devices can be simulated and analyzed, their responses under different ESD conditions can be simulated, and integrated circuits (IC) circuit optimization design can be carried out, thus saving the time and cost of testing [7].

Previous researchers have conducted a large amount of research on the ESD effect of IC and accumulated a lot of ESD protection circuit design and simulation experience [8,9]. At present, the commonly used circuit modeling schemes are the black box model based on chip port signal and the SPICE model of devices [10,11,12]. Although these models can describe the DC and AC characteristics of the circuit, it is difficult to describe the ESD effect of the device unless the model parameters are extracted by special experimental tests [13]. Technology computer-aided design (TCAD) can effectively describe the transient large-signal characteristics of components or the degradation characteristics under extreme conditions according to the physical characteristics of semiconductor devices, and it is generally applied based on physical principles [9,14]. With the development of computer technology, the calculation level of TCAD is improved, which can solve complex numerical models [15,16,17]. However, circuit-level TCAD modeling is rarely reported [18]. It is well known that the response of devices to electrical signals is also affected by their interconnected devices. So it is necessary to simulate and analyze the circuit level of on-chip ESD protection design.

In this paper, based on the 0.25 μm technology node, the ESD protection circuit numerical simulation model with multiple physical models is presented. Additionally, the negative transmission line pulse (TLP) injection simulation and damage experiment are carried out to explore the failure mechanisms of the ESD protection circuit. Section 2 describes the structure of the ESD protection circuit and its electrical characteristics. In Section 3, with the help of TCAD simulation software, it is studied that the interference and damage effect and mechanism of the ESD protection circuit are caused by TLP injection. In Section 4, the experimental process of the negative TLP pulse injection into the CD4069UBC chip is introduced, and the results are analyzed. Finally, the conclusions of this study are drawn in Section 5.

## 2. ESD Protection Circuit Model

### 2.1. The Structure of Circuits and Devices

As shown in Figure 1a, the study is based on typical ESD protection circuits in digital circuits. The circuit model consists of two diode TCAD models and one inverter TCAD model port cascade. In Figure 1b, the diode TCAD model is the physical equivalent numerical model of the three-dimensional shallow trench isolation (STI) diode. In the diode model, the anode is connected to the P-type heavy doping region and the cathode to the N-type heavy doping region. Moreover, the length of the electrodes is 0.35 μm. The double cathode diode structure is used because increasing the cathode width can reduce the conduction resistance and the over-surge voltage to improve DC performance and fast transient performance [19]. In addition, the three-dimensional diode model is extended along the *Z*-axis using the two-dimensional diode model, which optimizes the simulation speed while guaranteeing the simulation results [20]. In Figure 1c, the INV TCAD model is a three-dimensional physical equivalent numerical model of multi-gate CMOS. Moreover, C1 and C2 are the cross sections of PMOS and NMOS tubes in the INV model, showing that the model is a dual-gate, dual-source, single-drain CMOS and showing the electrode connections as inverters. C3 is a cross-section of the INV model’s source–drain channel, showing the dimensions of the active region and NWell, which are adjusted to make the model universally applicable. The gate oxide thickness of the CMOS is 0.015 μm, and the gate length is 0.35 μm. In the circuit, the gates of CMOS, the anode of D1, and the cathode of D2 are connected to the node “In”, and the drains of CMOS are connected to the node “Out”. The sources of positive channel Metal Oxide Semiconductor (PMOS) and the cathode of D1 are connected to the power supply V_SS_ = 5 V. The sources of Negative channel-Metal-Oxide-Semiconductor (NMOS) and the anode of D2 are connected to the ground. The initial temperature of all devices in the circuit is set to 300 K. Moreover, The thermal electrode is specified at the bottom of the devices, where the lattice temperature maintains at 300 K.

### 2.2. Physical Models Involved in the Model

In this paper, the equivalent physical model of the diode and the CMOS inverter is calculated based on the Poisson equation and continuity equation [21].
(1)$$\frac{\partial n}{\partial t}=\frac{1}{q}\nabla \cdot J_{n}-U_{n}$$
(2)$$\frac{\partial n}{\partial t}=\frac{1}{q}\nabla \cdot J_{p}-U_{p}$$
where *n*(*q*) is the concentration of electrons(holes), q represents the absolute value of the amount of charge carried by an electron, ***J***_n_(***J***_p_) is the current density vector of electrons(holes), *U*_n_(*U*_p_) is the net recombination rate of electrons(holes) inside the device. Moreover, the SRH recombination model, Auger recombination model, and avalanche ionization model are considered in the net recombination rate.
(3)$$J_{n}=qD_{n}\nabla n+nq\mu_{n}\left(\nabla \phi_{n}+P_{n}\nabla T\right)$$
(4)$$J_{p}=qD_{p}\nabla p+pq\mu_{p}\left(\nabla \phi_{p}+P_{p}\nabla T\right)$$
where *D*_n_(*D*_p_) is the diffusion coefficients of electrons(holes), *µ*_n_(*µ*_p_) is the mobility of electrons (holes) considering the high field saturation model, and *P*_n_(*P*_p_) is the thermal power of electrons (holes) derived from the thermoelectric power analytic-TEP model. The mobility is correlated with the doping concentration, and the high field velocity saturation effect is considered. This makes the model in this paper can accurately simulate the changes in electrical and material properties under the high temperature of the strong field. In addition, for the CMOS inverter circuit, a tunneling model of the gate oxide layer is added to the gate oxygen contact surface, which can better reflect the gate leakage current in the experiment [22].

### 2.3. Electrical Characteristics Simulation of the Model

The IV characteristics of the diode model are shown in Figure 2. The left side of Figure 2 describes the reverse bias characteristics of the diode model, which is described by the blue part, and its reverse breakdown voltage is 27.24 V. The right side of Figure 2 describes the positive bias characteristics of the diode model, which is described by the red part. Its positive conduction voltage is 0.7 V, and the positive transition voltage is 1.87 V.

The static input and output characteristics of the CMOS inverter circuit are shown in Figure 3, where *V*_OH_ = 2.956 V, *V*_M_ = 1.502 V, *V*_OL_ = 0.288 V, *V*_IL_ = 1.690 V, *V*_IH_ = 2.766 V. 

## 3. Simulation and Result Discuss

TLP is a common ESD test scheme [3]. In this paper, a single rectangular current signal is used to simulate TLP injection and analyze the failure mechanism of the ESD protection circuit. Based on the above physical equivalent numerical model, the simulation circuit is built, as shown in Figure 4.

### 3.1. Transient Simulation Analysis of the Model

Based on previous studies and the actual TLP pulse signal, the negative TLP simulated input signal is a rectangular wave current signal [23,24]. In Figure 4, the TLP pulse generator is partially equivalent to a current source Is and a resistance Rs. Moreover, D1, D2, and INV constitute the ESD protection circuit model. The response of the circuit model to TLP injection can be studied by observing the voltage and current of the nodes In and Out. Figure 5 shows the transient signals of nodes In and Out when negative polar TLP pulses are injected. In Figure 5a, the TLP negative polar current signal has a rising edge of 1ns, a pulse width of 5 ns, an amplitude of 0.13 A, and a decreasing edge of 1 ns. As shown in Figure 5b, the pulse signal begins to inject at time = 0.5 ns, and V_In_ = 0 V. So the ESD protection circuit is not started when the input resistance R_In_ at node In is much greater than R_S_ = 50 Ω, which makes the V_In_ rise rapidly and approximately equal to V_S_. The ESD protection circuit then starts, and the current is discharged from the D2 to the ground. Meanwhile, the input resistance R_In_ decreases rapidly, approaching R_S_. As shown in Figure 5a, the I_D2_ is approximately equal to the I_S_ during TLP, so the D2 plays a protective role during negative polarity injection so that the gate of the CMOS inverter does not break down. However, a large amount of current passes through D2, which also results in a large amount of Joule heat inside the D2 tube, causing its temperature to produce a high-temperature area, which may cause burnout damage. Moreover, due to the existence of the gate capacitance, the I_INV_ changes with the INV input voltage V_In_, which makes the output V_Out_ of the CMOS inverter change in the same direction as the V_In_. If the peak burr of the output signal changes greatly, it may cause the output to flip. To meet the design requirements for chip testability, a test interface can be added at the output of the inverter to detect if ESD can cause bit errors in the digital circuit [25,26].

And the internal peak temperature of CMOS INV only fluctuates slightly at the beginning of the TLP injection, as shown in Figure 6. Because the V_In_-V_SS_ value is less than the D1 reverse breakdown voltage, it is in the reverse cut-off state, and the D1 temperature is almost unchanged.

### 3.2. Mechanism Analysis of Burning Damage

In the simulation, when the internal temperature of the diode or inverter reaches the melting point of the material, the ESD protection circuit is judged to be damaged. In this paper, the failure of the ESD protection circuit when TLP injects a negative current ESD stress pulse [27] is simulated and studied. During the current ESD stress pulse injection, D2 is prone to high-temperature combustion, while D1 and INV have no obvious temperature change. As shown in Figure 7, the discounted diagram describes the change of internal maximum temperature with time. When I_S_ = 0.13 A, the maximum temperature inside the D2 failed to reach the melting point of the material within the pulse duration, and then with the disappearance of the pulse, the temperature of the D2 decreased to determine that there was no burning damage. When I_S_ = 0.14 A, with the pulse injection D2, the internal temperature continues to increase, and the final D2 internal maximum temperature reaches the melting point of the material, determining the D2 burn damage. With the increase of signal amplitude *V*_top_, the energy absorbed by the device increases at the same time to achieve the energy required for thermal breakdown faster, resulting in the reduction of device damage time.

Taking I_S_ = 0.23 A as an example, the distribution area of multiple physical parameters at D2 damage time is extracted, and the cross-section along the *Z*-axis is shown in Figure 8. As shown in Figure 8a, there is a high-temperature area represented by red in the anode contact of D2. Under the simulation conditions, it is determined that the anode of D2 is prone to burn damage. Extract the current density distribution and electric field intensity distribution of D2 at the time of burning, and the local image of their proximity to the electrode region is shown in Figure 8b,c. In Figure 8b, the closer to the electrode contact position, the greater the current density, while the D2 is a P^+^/N^−^/N^+^ structure with a single anode and double cathode, and all of the electrode contacts are 0.35 μm, which leads to the maximum current density at the anode. Moreover, it is easy to produce a large amount of Joule heat, resulting in the burning of the anode contact position. The ESD damage type of the circuit model is energy damage. Therefore, increasing the anode contact area can reduce the current density near the anode and reduce the generation of anode Joule heat. In Figure 8c, the anode and cathode are isolated by the STI oxidation area; it is easy to have a large, strong field intensity area at the bottom of the oxidation area, especially near the area with small curvature of two corners at the bottom. Combined with the Yellow area in Figure 8a, it is judged that D2 is also easy to burn down due to the large electric field near the bottom two corners of the STI oxidation area.

## 4. TLP Experiment and Result Analysis

### 4.1. TLP Experiment Process

In this paper, CD4069UBC is selected as the experimental sample for the TLP injection damage experiment. CD4069UBC is an inverter chip with diode ESD protection. And it is a product of Fairchild Semiconductor, a Texas company. Figure 9 shows the TLP pulse effect experimental injection platform, mainly including [25,26]: E3649A DC source used to supply power to the device under test (DUT);System PC used to complete TLP pulse parameter setting, data storage, and other operations;A TLP pulse generator module for generating the required TLP pulse signal and a small signal to detect leakage current;The peak injection pulse collected by the TDS3054C oscilloscope;The MDO3102 oscilloscope is used to record the response voltage waveform at the output port of DUT monitoring.

The cascade of two-stage inverters is used as the sample to be tested. The first stage inverter is used to study the TLP injection damage effect, and the output of the second stage inverter is connected with the oscilloscope to determine whether the DUT function is invalid and protect the oscilloscope. The TLP test system is set as follows: the negative polarity voltage injection range is 0 V~1100 V, the voltage step is 5 V, the pulse interval is 1000 ms, and the system leakage current failure tolerance is 0.5 A. The DUT is powered by a 5 V DC source, and its GND port is grounded. When the leakage current of the DUT increases suddenly, it is determined that the DUT is damaged. Then the function of the inverter is detected with a 5 V square wave signal to prove its function invalidation.

### 4.2. The Result Analysis

Figure 10 is the damaged DUT after TLP injection. Moreover, Pin1 is the DUT input port In, and Pin2 is the DUT output port Out. The area of the yellow box is the diodes of the ESD protection circuit. It can be seen that there are obvious high-temperature burn marks at the diodes, indicating that the chip has been burned and damaged due to TLP injection, and the TLP pulse causes a large current to flow through the circuit. In the yellow box in Figure 10, there is a double cathode diode, consistent with the diode model in Figure 8, connecting the input port to the ground port, and the diode has obvious burn marks. However, in the blue box area in Figure 10, the multi-gate inverter is in good condition. The experimental results are in agreement with the simulation results, reflecting the authenticity of the simulation model.

## 5. Conclusions

In this paper, a physical equivalent numerical model of the ESD protection circuit is proposed, and transient simulation and corresponding TLP injection experiments are carried out. The negative polarity TLP injection experiment is simulated by injecting negative polarity single rectangular wave current signals with different amplitudes at the input of the ESD protection circuit model. Based on TCAD simulation, we extract the input and output transient signals of the circuit model, and the physical principle of signal change is analyzed. The simulation results of TLP injection show that a negative polarity pulse can cause burrs in the high-level output signal and can easily cause high-temperature damage to the diode D2 between the input end and the grounding end in the circuit model. Moreover, the damage time decreases with the increase of the peak value of the injected pulse current. In addition, by extracting the temperature distribution, current density distribution, and electric field intensity distribution of D2, we found that Joule heat caused by the high current is the mechanism of diode damage. In the *CD4069UBCN* chip TLP injection damage experiment, it is found that the diode between the chip input and the grounding is burned, which is consistent with the simulation results, which proves that the simulation method is feasible.

The ESD protection circuit model in this paper is authentic, which can help the ESD reliability analysis of the IC. Moreover, the modeling and simulation analysis method in this paper is scalable and supports the current ESD stress pulse analysis of the CMOS circuit.

## Figures and Tables

**Figure 1 micromachines-14-00600-f001:**
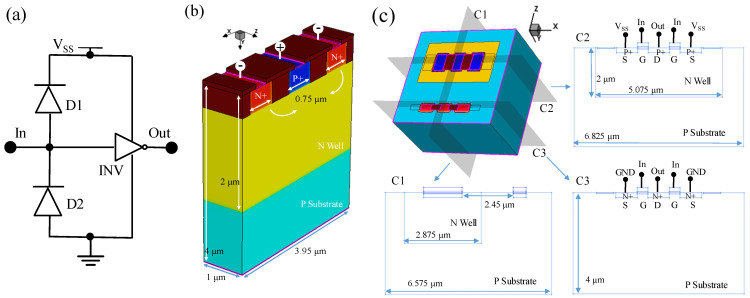
(**a**) The ESD protection circuit diagram. (**b**) The TCAD model of the STI diode. (**c**) The TCAD model of the CMOS inverter and its cross sections.

**Figure 2 micromachines-14-00600-f002:**
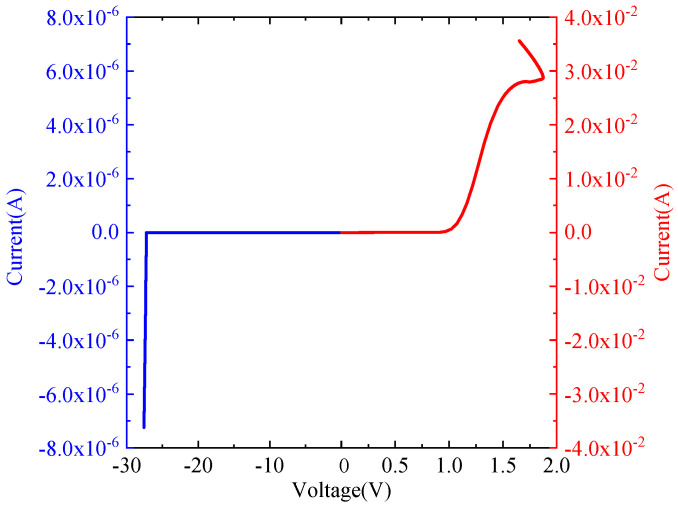
The IV characteristics of the diode model.

**Figure 3 micromachines-14-00600-f003:**
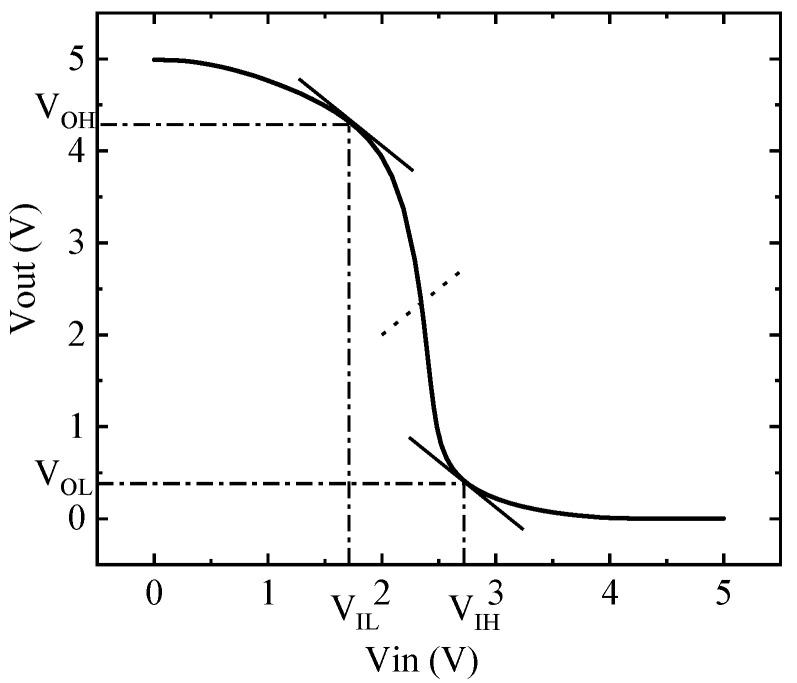
The static voltage characteristics of the inverter model.

**Figure 4 micromachines-14-00600-f004:**
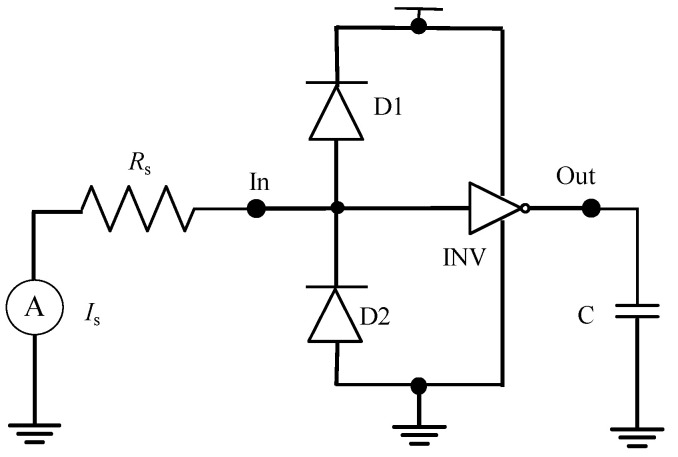
The ESD protection circuit model simulation schematic.

**Figure 5 micromachines-14-00600-f005:**
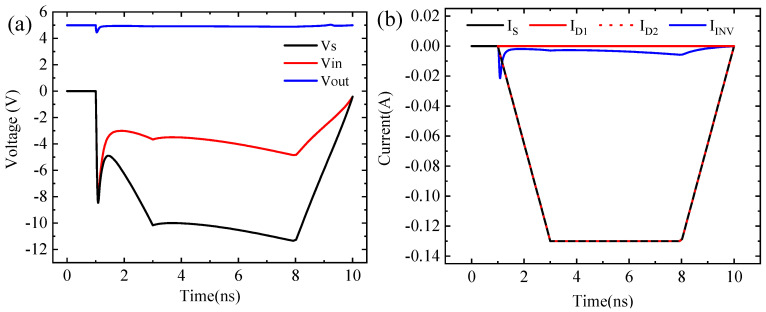
The transient signals of nodes In and Out. (**a**) Variations of the circuit node voltages with time; (**b**) Variations of the circuit current with time.

**Figure 6 micromachines-14-00600-f006:**
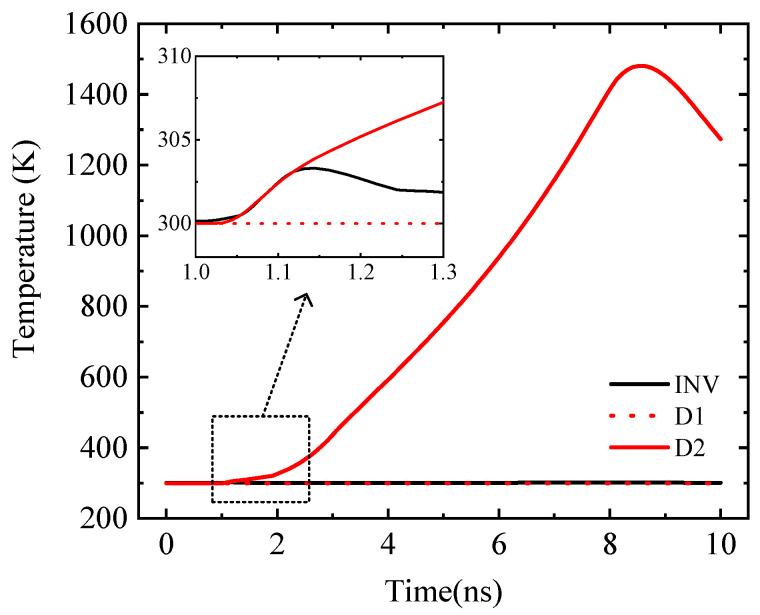
Devices’ peak temperature change curve with time.

**Figure 7 micromachines-14-00600-f007:**
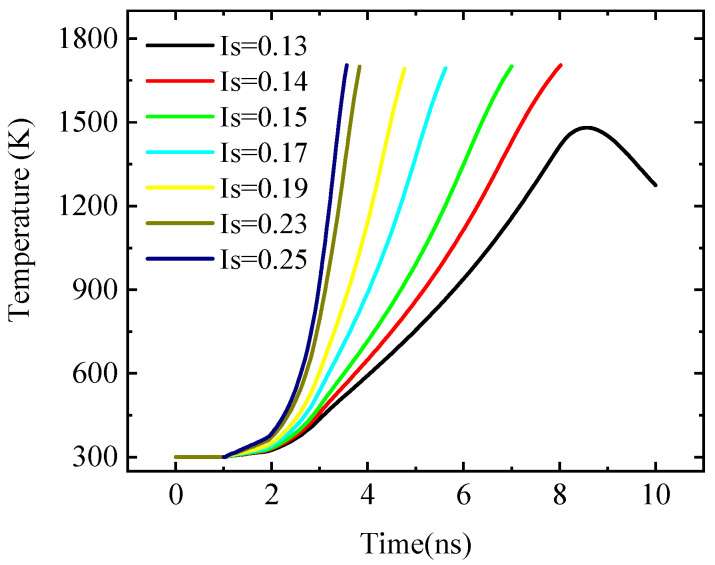
Variation of the internal maximum temperature of the D2 over time.

**Figure 8 micromachines-14-00600-f008:**
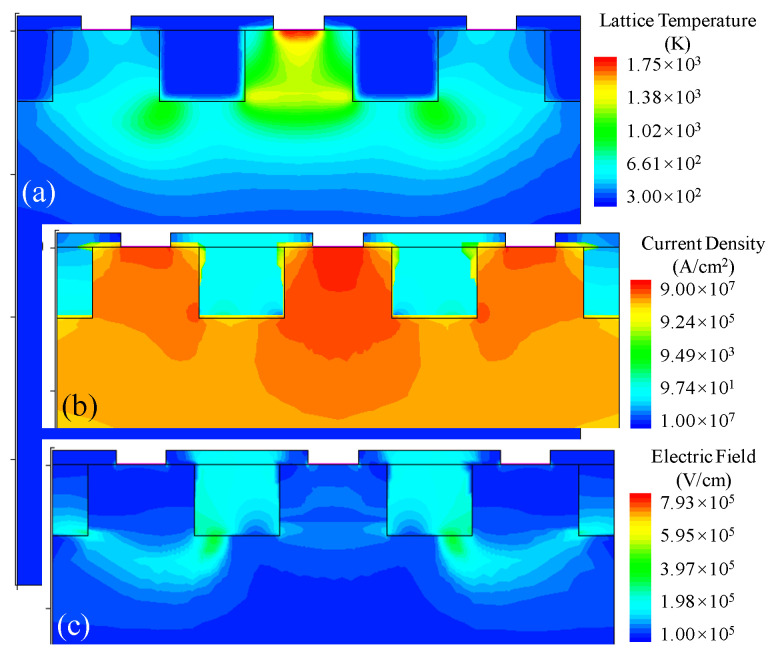
(**a**) The distribution of temperature (K), (**b**) The distribution of current density (A/cm^2^), (**c**) The distribution of temperature (V/cm).

**Figure 9 micromachines-14-00600-f009:**
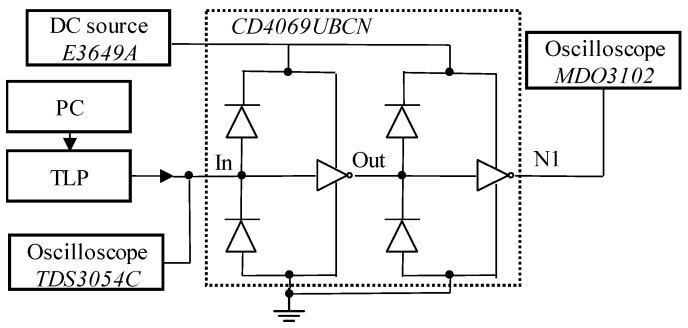
The TLP test platform.

**Figure 10 micromachines-14-00600-f010:**
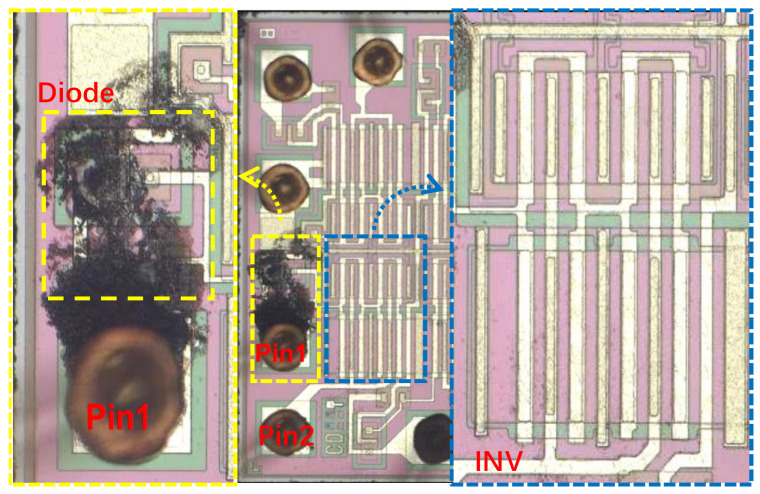
The ESD protection circuit model simulation schematic.

## Data Availability

Not applicable.

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
