# Peer review of "Study on ESD Protection Circuit by TCAD Simulation and TLP Experiment"

_micromachines, 2023, doi:10.3390/mi14030600_

Round 1

Reviewer 1 Report

Journal: micromachines

Manuscript ID: 2203326-peer-review-v1

Title :Study on ESD Protection Circuit by TCAD Simulation and TLP Experiment»

type: regular

Complete List of Authors:

Fuxing Li 1*, Changchun Chai 1, Yuqian Liu 1, Yanxing Song 1, Lei Wang 1 and Yintang Yang 1

Thanks for Authors for this study and paper proposal.

At glance, the paper introduces an electro-thermal TCAD simulations and TLP measurement on ESD protection in CMOS technology.  Find below some feedback to reinforce the impact of this article:

-          First of all,  in the abstract indicate the technology node used for this study (remind it in the introduction) and precise that the study is focused on negative TLP stress.

-          For the Design and devices : it seams that the ESD clamp is missing in the study or precise the main objective without explicit clamp.

-          do the inverter play the role of the ESD clamp  (precise)?

-          Or parasitic devices between diodes are the ESD clamp  (precise)?

-          If it is the case, give the TLP response of the inverter (Vss Vs Gnd)  or parasitic devices into diodes.

-          Do the diodes are isolated?

-          For TCAD indicate the Poisson’s equation and the heat equation used for the numerical study.

-          What are the thermal boundary conditions (Rth, Cth)?

-          What are the initial conditions (V,Tinit , Tbound ..)?

-          What are the sizes of diodes and inverter? Indicate too the gate oxide thickness .

-          The Standard TLP is 10ns rise time and 80 ns hold for 10ns fall down. Please justify the 1ns , 5ns & 1ns shape for the TCAD simulations which is not in the standard?

-          Indicate in Fig.5 that it is only for NEGATIVE pulse .

-          Why in sub-section 3.2 Authors talk about ELECTROMAGNETIC pulse , rather than current ESD stress Pulse ( see for example Ref : “In-depth Electromagnetic Analysis of ESD Protection for Advanced CMOS Technology during Fast Transient and High-Current Surge. IEEE Transactions on Electron Devices, v 61, n 6, p 1900-6, June 2014 ) ?

-           Fig 10  give the silicon de-processing to demonstrate the silicon signature damage and link to figure 8.

-          Why use multi level TLP pulses (several numerical runs) rather than fast ACS method (1 numerical run. ref :« Numerical evaluation between transmission line pulse (TLP) and average current slope (ACS) of a submicron gg-nMOS transistor under electrostatic discharge (ESD) », Workshop EOS/ESD/EMI, LAAS-CNRS Toulouse, 2002).

Thank you in advance for your time and efforts to improve this proposal with the consideration of the requested corrections.

BR

Author Response

Dear Reviewers:

Thanks for your letter and comments concerning our manuscript entitled “Study on ESD Protection Circuit by TCAD Simulation and TLP Experiment”. Those comments are all valuable and very helpful for revising and improving our paper, as well as important guiding significance to the researches. We have studied comments carefully and have made corrections which we hope meet with approval.The main corrections to this article and responses to the reviewer's comments can be found in the document "Reviewer1".

Reviewer 2 Report

This paper described a simulation process of an ESD protection scheme and described the model used. Chip characterization was also performed to validate the feasibility of the model and simulation. The authors told a good case study story but I just do not see any novelty in this paper, especially the ESD protection scheme that the authors tried to demonstrate, and there is not much value in current claims. This methodology has been well-known in the industry. I would like to see how authors could clarify their novelty compared to prior arts before we talk about technical details. English writing also needs further improvement.

Author Response

Dear Reviewers:

Thanks for your letter and comments concerning our manuscript entitled “Study on ESD Protection Circuit by TCAD Simulation and TLP Experiment”. Those comments are all valuable and very helpful for revising and improving our paper, as well as important guiding significance to the research. We have studied the comments carefully and have made corrections which we hope meet with approval.

The main corrections to this article and responses to the reviewer's comments can be found in the document "Reviewer2".
